# The Dynamics of Stand Structure Development and Natural Regeneration of Common Beech (*Fagus sylvatica* L.) in Plitvice Lakes National Park

Tomislav Dubravac [1], Damir Barčić [2,*], Roman Rosavec [2], Željko Španjol [2] and Sead Vojniković [3]

1   Croatian Forestry Institute, 10450 Jastrebarsko, Croatia; tomod@sumins.hr
2   Faculty of Forestry and Wood Technology, University of Zagreb, 10000 Zagreb, Croatia; rrosavec@sumfak.hr (R.R.); zspanjol@sumfak.hr (Ž.Š.)
3   Faculty of Forestry, University of Sarajevo, 71000 Sarajevo, Bosnia and Herzegovina; s.vojnikovic@sfsa.unsa.ba
*   Correspondence: dbarcic@sumfak.hr

**Abstract:** The authors investigate the structural characteristics, regeneration processes, growth, development, and survival of a young generation of common beech (*Fagus sylvatica* L.) based on three periodic measurements (1998, 2009, and 2018). The studied forest community (*Lamio orvale-Fagetum sylvaticae* (Ht. 1938) Bohridi 1963) is situated within a forest reserve in Plitvice Lakes National Park, Croatia. Monitoring was carried out according to UN/ECE (2000) for systematic and repeated research. The basic structural indicators, structural canopy elements, and appearance of the young generation were measured as the basic conditions of natural restoration in repeated phytocenological surveys (1980, 1988, 2004). The distribution of the number of trees of the first two measurements (1998–2009) indicates a distribution with pronounced right asymmetry. The structural dynamics observed during the surveys and alongside vegetation surveys indicate the dynamics of the growth and development of beech. The results show that the main drivers of development dynamics in protected forest ecosystems are structural breaks (gaps) that occur in stands due to the action of abiotic and biotic factors. The passive management model in effect in the national park should be replaced with a more active approach to facilitate natural processes with the aim of preserving and renewing the forest ecosystem.

**Keywords:** monitoring; natural regeneration; forest habitat; protected area; climatic vegetation

## 1. Introduction

Common beech (*Fagus sylvatica* L.) is the most widely distributed tree species in the Republic of Croatia in terms of surface area and wood stock. Beech forests in Croatia have a natural structure and natural distribution range, with good natural regeneration, indicators of good management and stand stability, biodiversity, and productivity. Accordingly, there are no beech plantations or artificial forest cultures. Given the current state of beech forests, they are highly valuable, and any degradation would have serious consequences on ecosystem stability, with significant economic and overall losses as a result. Fortunately, beech forests are in relatively good health, with good resilience to environmental pollution in comparison with common fir and pedunculate oak, which are more susceptible to sudden ecological changes [1].

One of the most important indicators of a forest's optimal condition and natural state is natural regeneration. Management (passive or active) in protected areas can also serve to improve the commercial value and the general ecological and social value of these forests while supporting optimal natural stand structure and lasting protection and development of forest soils and habitats. Natural regeneration of forests has become burdened with negative ecological (particularly climatic) changes in forest ecosystems, with subjective omissions in management also a contributing factor. Therefore, the forestry industry needs

to seek out and find solutions that could mitigate the consequences for the forest ecosystem. In commercial forests, every forest ecosystem should be directed towards progressive development that gives maximum production with stability and secured natural regeneration. In protected areas (particularly national parks), forest ecosystems should also be treated according to the same principles of forestry science while respecting the influence of natural structure, conserving biodiversity, and preserving forest services (hydrological, erosion control, etc.) that dictate the guidelines for management plans. Protected forest ecosystems, such as those in national parks, are significant for the development of primary and applied natural sciences, and their research is imperative for the assessment of effective forest management [2]. This has an impact on the stability and preservation of forest ecosystems and has provided the opportunity for the establishment of national parks in the mid to late 20th century in which forests are the primary or one of the main fundamental phenomena protected in the park.

Virgin and old-growth forests are an important basis for studying how forest ecosystems grow and develop. Though areas classified as primary forests are well-represented in the world (35.7% of forested areas), this figure is low in Europe (just 2.8% excluding Russia) [3]. Through Europe's long history, nearly no areas have retained their unchanged natural character [4,5], and several researchers have emphasised the importance of understanding such ecosystems [6–9]. There is an increasing trend towards managing forests on an ecological basis, and it is therefore essential to transfer the understanding of natural stand processes into practice. A number of studies to date have examined stands of old-growth beech forests [2,8–12], and most have taken a traditional approach to describing structural properties and mapping development phases. Newer research is taking an approach to understanding the effects of natural canopy gaps on forest development dynamics [13–19], while others have applied time series data [20–22].

The present study is a continuation of systematic, multiyear research conducted to gather data on a permanent experimental plot in a hilly beech forest with dead nettle (*Lamio orvale-Fagetum sylvaticae* Ht. 1938) in the Medvjeđak Forest Reserve within Plitvice Lakes National Park, which is subject to conditions of passive protection [19]. The aim of this study was to determine the state of the stand structure, the development of the canopy structure, regeneration processes, the properties of new generations, and the conditions for its development and survival given the inner vertical and horizontal stand structure.

Research to date has examined the fundamental structural characteristics, properties of stand growth and canopy development dynamics, and the abundance and quality of the new generation, and the results have indicated a degraded stand structure. Furthermore, an important part of the research is focused on satisfactory natural regeneration within national parks, because under conditions of passive protection this regeneration is questionable and less than satisfactory, particularly when taking into account the increasingly frequent climatic anomalies over the past 30 years [23–27]. This is indeed a matter of importance for all European forests. Europe's forested area can be distinguished into forest areas available for sustainable harvesting and those protected forests not available for harvesting, e.g., national parks and nature conservation [28,29]. According to the State of Europe's Forests 2020 report [30], about 50 million ha, or 23.6% of the forests in Europe, are in protected areas. In most of these areas, limited harvesting is allowed if other non-wood services are secured. In that sense, Croatia should also work towards ensuring the stability of its forest ecosystems and their regeneration by permitting limited management to take place within all protected areas, with the exception of strict and special reserves. This research aims to support this claim.

## 2. Materials and Methods

### 2.1. Study Area

Beech forests with dead nettle are distributed throughout the Dinaric area of Gorski Kotar, Mala Kapela, and Velika Kapela, Velebit, in the Plitvice Lakes area and in northwestern Croatia (Samoborsko Gorje, Strahinščica, Ivanščica, Medvednica, Moslavačka Gora,

and Kalnik), which largely overlaps with the distribution range of the species *Lamium orvala*. This community is distributed at elevations from 400 to 800 m, on varying expositions, flat terrain, plateaus, smaller ridges, and gentle slopes [31]. In the Dinaric karst areas, the dominant substrates are Mesozoic limestone and Dolomites with some presence of silicate and silicate–carbonate clastic rock, with magmatic rock present only locally. In this area, this community develops primarily on brown soil and black soil on limestone substrate, and less often on dolomite and red soil substrates.

Beech forests in Croatia encompass all of the climate zones, according to the Köppen classification, that are present in Croatia [32]. As a pronounced mesophyte, common beech is best suited to areas with moderately warm summers, a large quantity of precipitation, and shorter winters. Considering temperature, common beech optimally develops in areas with an average annual air temperature of 7 to 10 °C. In the southern part of the distribution range of the community *Lamio orvale-Fagetum*, the average air temperature is about 8 °C, with a mean annual precipitation of about 1700 mm [32], while at the northern end of the range, the average annual air temperature is 9.5 °C, with mean annual precipitation of 1100 mm.

The northeastern part of Plitvice Lakes National Park is primarily covered with mountain beech forests (*Lamio orvale-Fagetum sylvaticae*) at an elevation of 700 to 900 m. Above them, in the northwest, southwest, and part of the southeast areas, we find mountain beech with fir forests (*Omphalodo-Fagetum*), while at elevations below 700 m (550–700 m), we find forests of sessile oak and common hornbeam (*Epimedio-Carpinetum betuli*). Common beech is the most widely distributed tree species (65%), followed by common fir (25%), common spruce (5%), and other tree species. Of the total forest area, 9500 ha are seed plants, 4000 ha are coppice forests, and the remainder is covered by various degradation stages of thickets [31,32].

Focused research has been conducted in the Medvjeđak Forest Reserve (Figure 1), which was established in 1976 and is situated within a large forest complex of beech forests in the northwestern park of Plitvice Lakes National Park (Figure 2). The reserve includes three sections covering a total surface area of 152 ha. The highest elevation within the reserve is 975 m, and the lowest is 580 m. Initial research within the reserve began in 1980 as part of the international UNESCO project entitled "Man and Biosphere", while vegetation mapping was conducted as part of the IUFRO programme [33,34].

## 2.2. Experimental Design

Experimental plots were set up with the aim of observing the dynamics of the forest ecosystems under a strict protection regime. The permanent experimental plot (1 ha in size, coordinates 44°53′09″ N; 15°38′01″ E) was established in 1998 according to [19]. Stand age at the time of establishment of the monitoring plot in the national park (1998) was 147 years. In the most homogenous part of the plot, a subplot (60 × 60 m) was set up (Figure 3). A LaserAce 300 laser measurement device was used to measure the elevation of terrain at the root collar of each plant, and at several additional, characteristic points of the terrain. The obtained data with corresponding $x$, $y$, and $z$ coordinates were analysed and interpolated with the program ESRI ArcMap to obtain a digital model of the terrain in the experimental plot (Figure 3A,B) [35]. Horizontal projections of the crown were also digitised in the ArcMap programme, and maps of soil shading by the canopy based on tree species and layers were created (stand visualisation, Figure 3C) using the program EnVision [36].

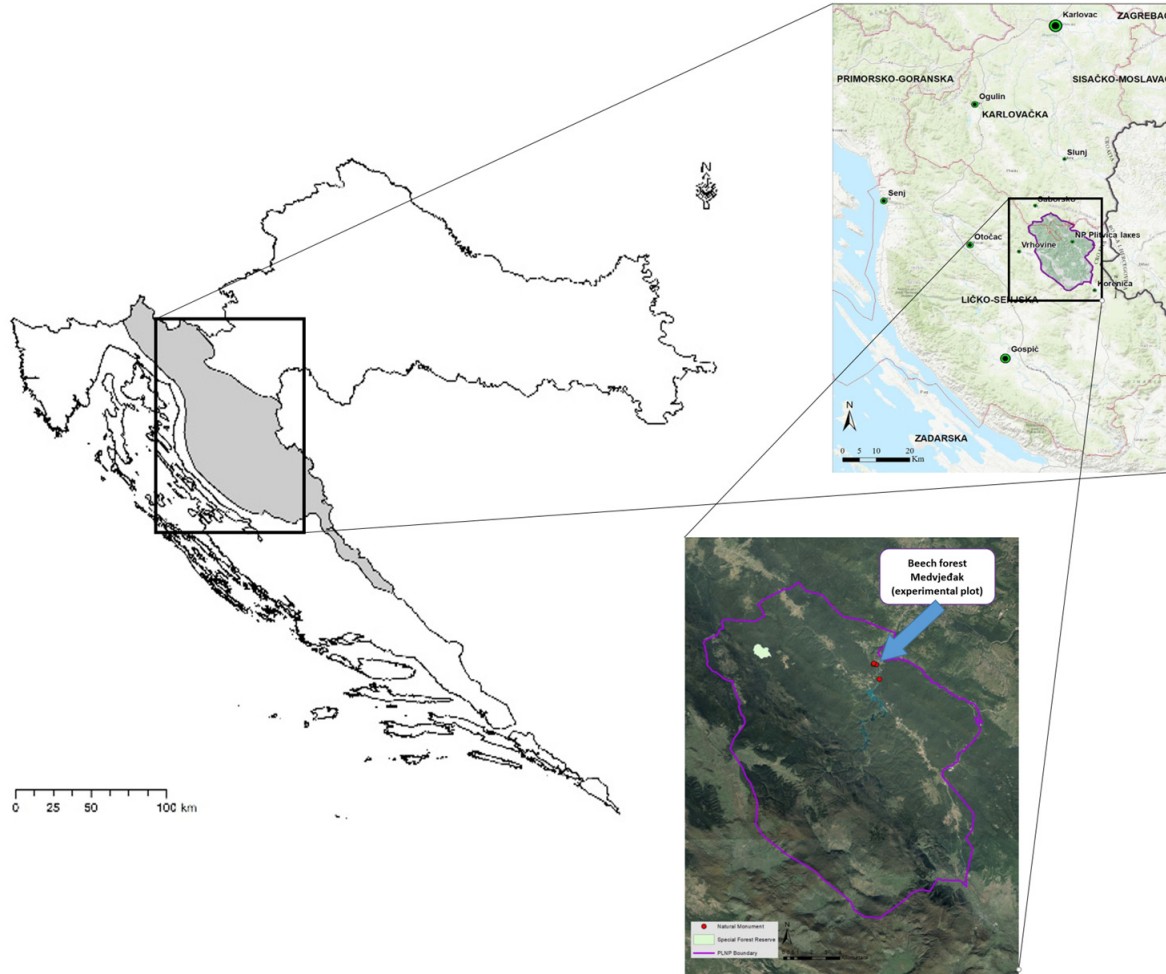

**Figure 1.** Plitvice Lakes National Park (area 29,685.15 ha) in the continental part of the Dinaric karst in Croatia, with the Medvjeđak old-growth beech forest reserve (area 152 ha) lying within the park boundaries.

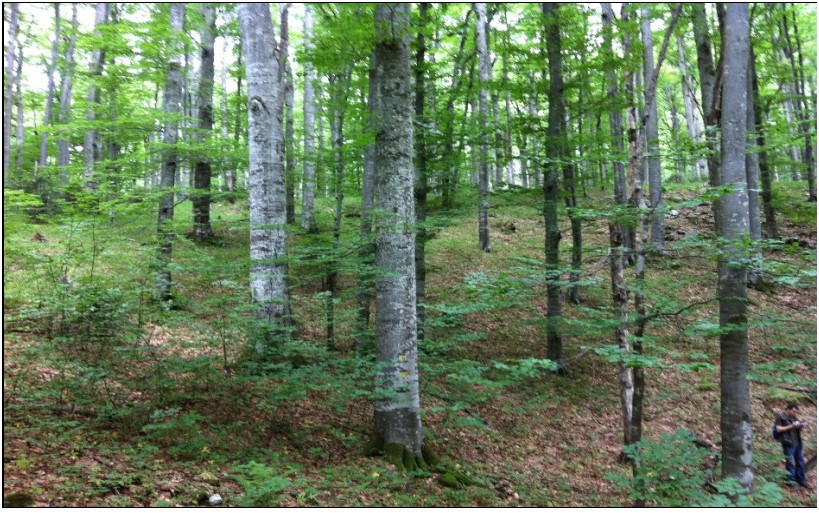

**Figure 2.** Association of *Lamio orvalae-Fagetum* in the Medvjeđak Forest Reserve within Plitvice Lakes National Park (Photo: T. Dubravac).

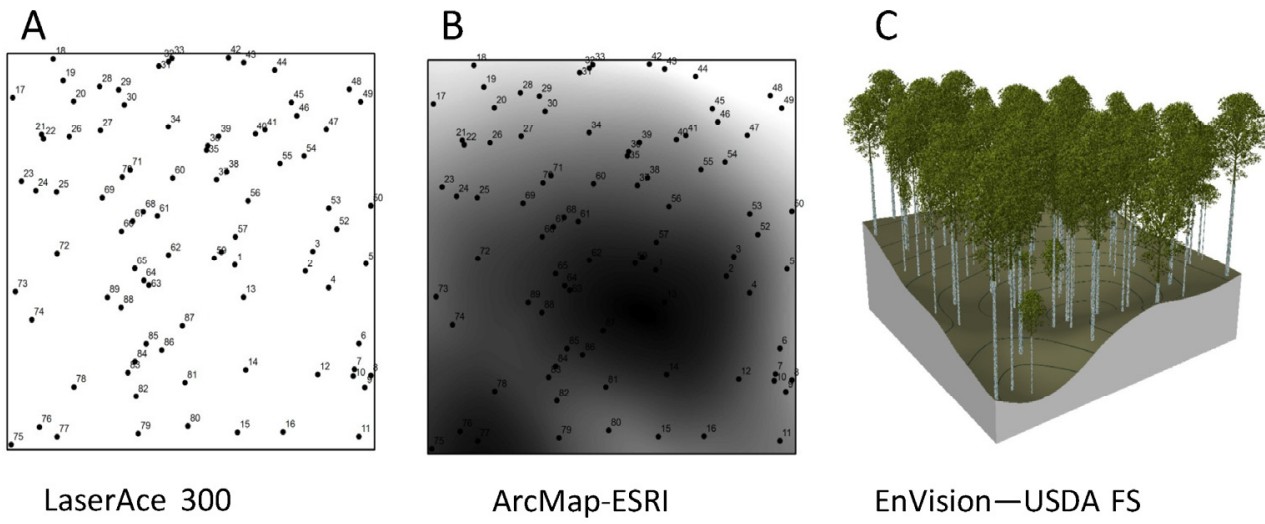

**Figure 3.** Subplot (dimension 60 × 60 m) and location of trees (**A**); digital model of the terrain (**B**); stand visualisation (**C**) according to Dubravac et al. [19].

Measurements of the plots were performed periodically at 10-year intervals in 1998, 2009, and 2018. Within the subplot (60 × 60 m, Figure 4), all trees with a diameter at breast height (DBH) greater than 7.5 cm were counted. The following measurements were made: (i) basic taxative data (DBH, tree height, length of trunk) were recorded; (ii) three transects (strips) 2 × 60 m were designated with a total area of 360 m², and on each transect we recorded the abundance and height structure of young trees and the shrub layer by height classes and species; (iii) dimensions of the new-generation trees, abundance, and species of new trees were recorded; (iv) during the last measurements of the new generation (2018), measurements were made of abundance of new trees, DBH, and height of trees; and (v) trees in the middle of the plot 20 × 60 m (stand profile) were recorded in the horizontal projection of the canopy for comparison with changes in the vertical profile of the stands between the two measurements (1998 and 2009). Digitisation of the horizontal projection of the canopy and development of the digital terrain map were performed using the ArcMap program v10.8.2, with data preparation and processing in MS Excel 2021 (Microsoft Corp., Redmond, WA, USA), while the programs Stand Visualisation System (SVS v3.36) and EnVision (v2.20, USDA Forest Service, Seattle, WA, USA) were used for stand visualisation and the vertical profile [36].

Research of the floral system included the creation of vegetation relevés on the permanent experimental plot in 1980, which was followed up in 1988 and 2004 using the standard Central European Zurich–Montpellier method [37,38]. Vegetation relevés included compiling a list of all plant species found within the tree, shrub, and undergrowth layers that were observed, with records of their values related to abundance and cover.

Plant species nomenclature follows the Flora Croatica database [39], while names of plant communities, their syntaxonomy position, and the sociological association of individual species are taken according to Vukelić [31].

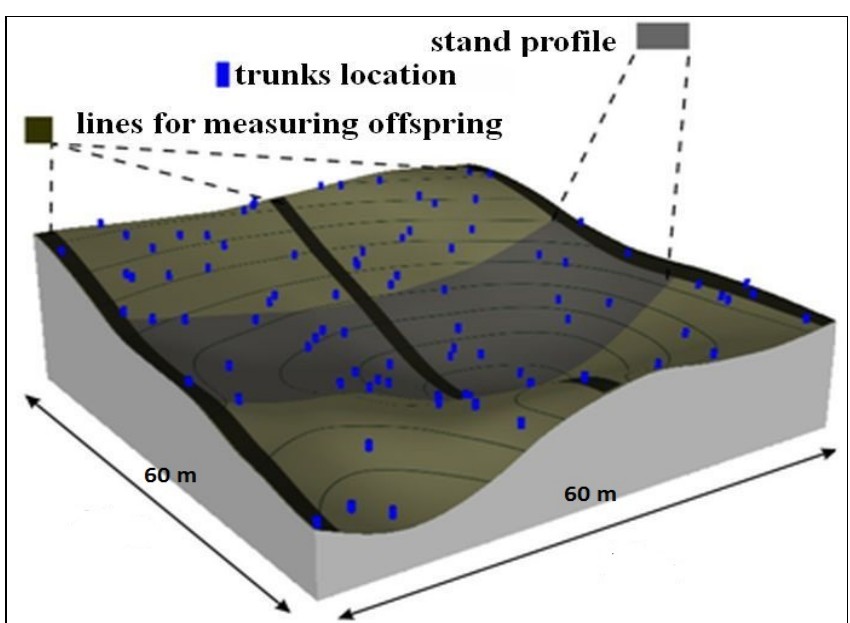

**Figure 4.** Digital terrain model (DTM) of the experimental plot with basic measuring elements according to Dubravac et al. [19].

## 3. Results

The basic structural stand characteristics (number of trees, basal area, and volume) on the experimental plot during measurements (1998/2009/2018) and the average values (DBH, height, basal area, and volume) are shown in Tables 1 and 2. Measurement data indicate that these are pure beech stands with a single age structure and group distribution. The distribution of tree numbers (Figure 5) in the first two measurements (1998–2009) indicates a normal (Gaussian) distribution of tree numbers with a pronounced right asymmetry that follows the relatively constant number of trees by DBH classes. Trees are grouped around the mean DBH (40 cm, Table 1). The most recent measurement interval (2018) shows a bimodal Gaussian distribution, with the first peak between 7.5 and 52.5 cm, and the second from 52.5 to 92.5 cm, with a significantly higher frequency of trees with a greater DBH. The number of trees in the second measurement decreased by 10 trees and by 105 trees at the final measurement. The largest reduction in tree numbers is in the smallest DBH classes up to 32.5 cm, while in the DBH class 37.5 cm, the number of trees was constant in all measurements, followed by a visible drop in tree numbers with DBH up to 52.5 cm. After this class, the number of trees in larger DBH classes increased. The reason for the loss of a number of trees was breakage of tree trunks due to wind or uprooting of trees, while over this 20-year period the largest portion of trees experienced a natural dieback (drying) caused by development stages in the stage (ageing). The distribution of volume by DBH classes shows a similar distribution (Figure 6).

**Table 1.** Basic structural characteristics.

| Measurement | N | G | V | DBH | h | g | v |
|---|---|---|---|---|---|---|---|
| Year | trees/ha | m²ha⁻¹ | m³/ha⁻¹ | cm | m | m² | m³ |
| 1998 | 301 | 43.11 | 656.41 | 39.3 | 26.9 | 0.14 | 2.18 |
| 2009 | 291 | 45.68 | 803.07 | 41.1 | 27.1 | 0.16 | 2.76 |
| 2018 | 196 | 37.80 | 695.80 | 46.8 | 32.0 | 0.19 | 3.55 |

Legend: (N—number of trees per hectare, G—stand basal area, V—stand volume and average values of tree characteristics, DBH—diameter at breast height, h—tree height, g—tree basal area, v—tree volume).

**Table 2.** Descriptive statistics.

| Measurement Years | 1998. | 2009. | 2018. |
|---|---|---|---|
| Mean | 39.10 | 41.11 | 46.83 |
| Standard Error | 0.928867 | 0.983053 | 2.09356 |
| Median | 38.00 | 40.50 | 44.30 |
| Mode | 37.00 | 35.50 | 39.75 |
| Standard Deviation | 16.12 | 16.77 | 17.64 |
| Sample Variance | 259.7007 | 281.2206 | 311.1924 |
| Kurtosis | 0.287316 | 0.141323 | −0.60844 |
| Skewness | 0.671723 | 0.587233 | 0.258718 |
| Range | 89 | 89.5 | 78.05 |
| Minimum | 11.00 | 11.00 | 10.90 |
| Maximum | 100.00 | 100.50 | 88.95 |
| Sum | 11768 | 11963 | 3324.853 |
| Count | 301 | 291 | 196 |

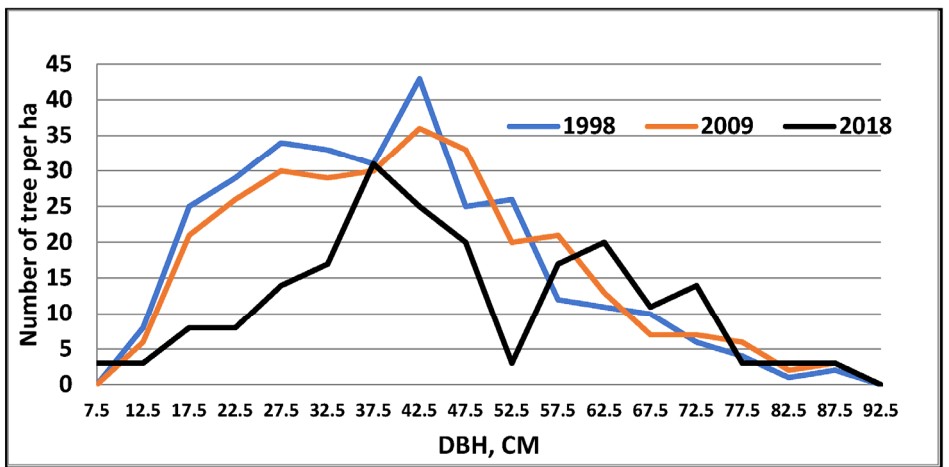

**Figure 5.** Distribution of the number of trees by DBH classes (frequencies of trees in 5 cm classes).

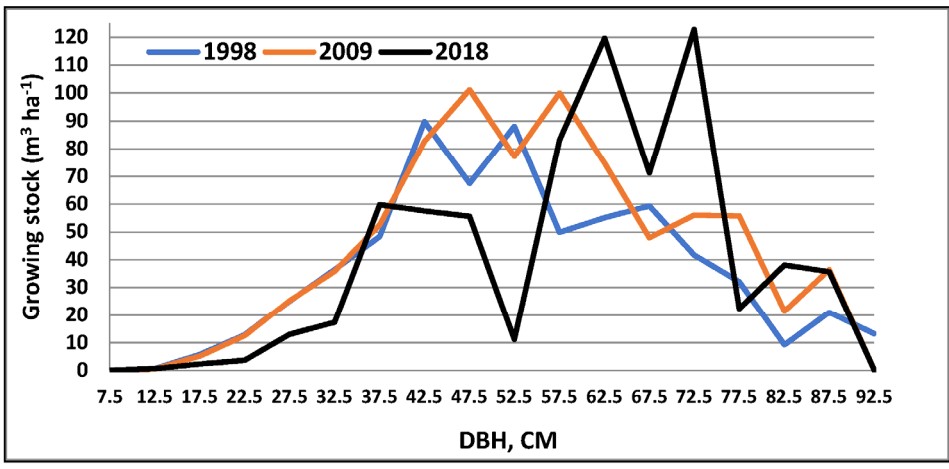

**Figure 6.** Distribution of volume by DBH classes (frequencies of trees in 5 cm classes).

Other structural elements, such as basal area and stand volume, were also monitored (Table 1). The total basal area of the first measurement (1998) was 43.11 m$^2$ha$^{-1}$ [19] and

in the second measurement (2009) it was 45.68 m$^2$ha$^{-1}$, for an increase of 2.57 m$^2$ha$^{-1}$, while in the third measurement the basal area was 37.80 m$^2$ha$^{-1}$, which is a decrease of 7.88 m$^2$ha$^{-1}$ from the second measurement and of 5.31 m$^2$ha$^{-1}$ from the first measurement. The total volume increased by 146.6 m$^3$/ha$^{-1}$, from 656.41 m$^3$/ha$^{-1}$ (1998) [19] to 803.07 m$^3$/ha$^{-1}$ (2009), and in the final measurement (2018), the total volume was 695.80 m$^3$/ha$^{-1}$, indicating a loss of volume by 107.27 m$^3$/ha$^{-1}$. These volume data in the three measurement intervals, as shown by DBH class (Figure 7), indicate how volume was accumulated in the old, mature trees of common beech (trees with a DBH greater than 51 cm). In examining the average values of DBH, height, basal area, and volume, an increasing trend is evident during the study period.

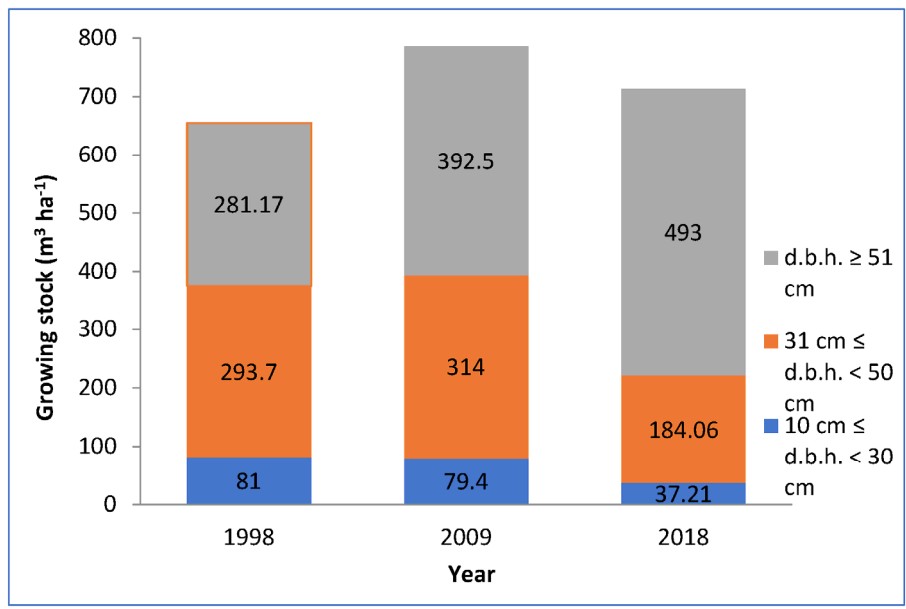

**Figure 7.** Distribution of volume in three DBH classes during measurements.

The horizontal canopy projections (Figure 8) in the first measurement (1998) found soil shading by the canopy of 96% [19]. The average surface of the horizontal crown projections was 53.67 m$^2$, with a range from 5.94 to 158.36 m$^2$. In the lower left quadrant of the experimental plot (Figure 8), an initial regeneration core was established in a canopy gap, created by the dieback of a dominant beech tree from the upper stand layer prior to establishment of the plot. From the DBH of that dead tree, the area of the horizontal crown projection was established at 145 m$^2$ [19]. The resulting gap in the canopy enabled the emergence of an initial regeneration core.

To assess the dynamics of crown structure development between the two measurements (1998–2009), two stand profiles were made for a section of the plot 20 × 60 m in size (Figure 9). The sum of the area of the horizontal crown projection of 27 trees in the profile in the first measurement was 1492.24 m$^2$, with a mean projection area per tree of 55.27 m$^2$ from 656.41 m$^3$/ha$^{-1}$ (1998) [19]. By the second measurement, three trees with a total horizontal crown projection area of 69.27 m$^2$ had died. However, there was an increase in the average crown projection per tree to 63.04 m$^2$ (+7.77 m$^2$) and an increase in the total horizontal crown projection area to 1512.89 m$^2$ (+20.65 m$^2$) from 656.41 m$^3$/ha$^{-1}$ (1998) [19]. In the period between the two measurements, there were no significant changes in the vertical profile, regardless of the loss of three trees.

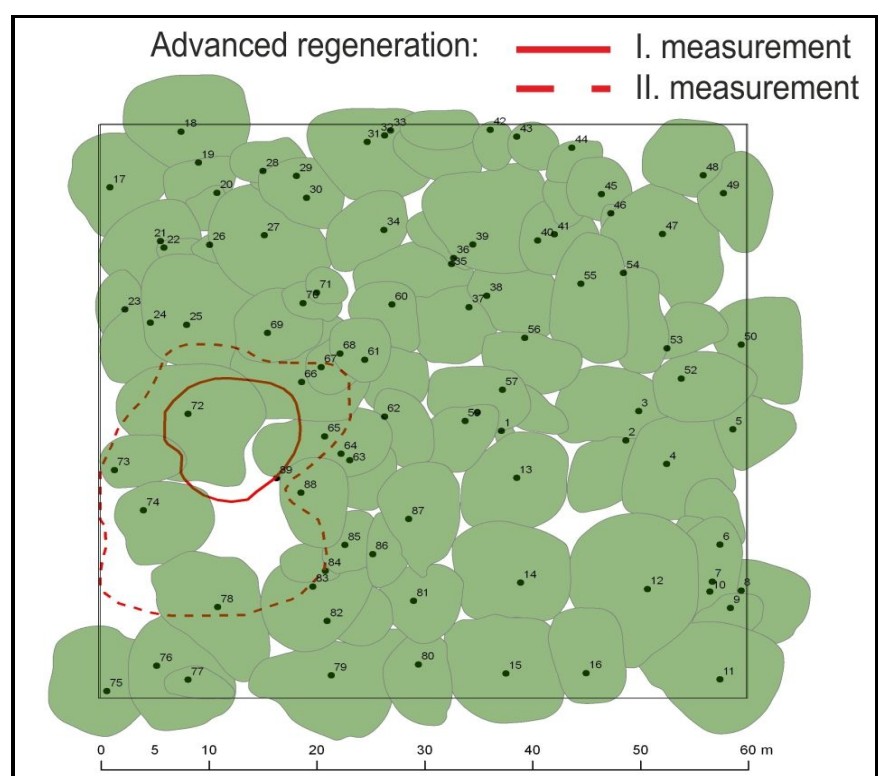

**Figure 8.** Horizontal crown projection with the initial cores and gaps, with first measurement in 1998 (solid line) and second measurement in 2009 (dashed line), according to Dubravac et al. [19].

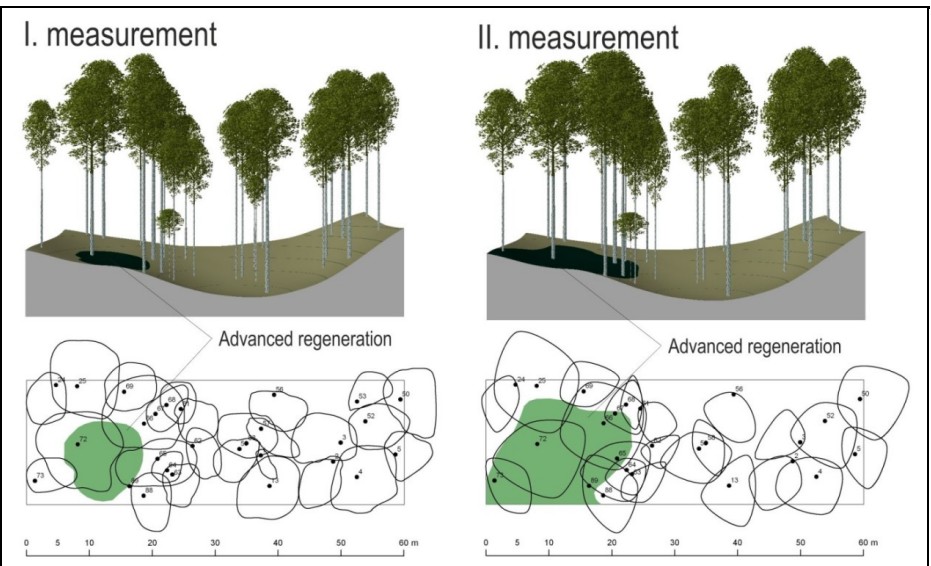

**Figure 9.** Gap dynamics in the Medvjeđak Forest Reserve in the study period (1998–2009) according to Dubravac et al. [19].

The vegetation relevés (Table 3) from 1980, 1988, and 2004 indicate very minor changes in the beech forest in the short period of eight years, though after 24 years, the trend indicates an increase in the shrub layer, which is associated with the appearance of gaps (Figures 8 and 9). With one important note on the images, there is a very low representation of common beech in the shrub and ground layers. The gaps suggest opportunities for more intensive natural regeneration, and the dominant tree species (common beech) is in correlation with this.

**Table 3.** Vegetation relevés on the experimental plot.

| Plot number: | TPP 31 | | | Soil: | | | Classification by Braun-Blanquet—in abundance and cover | | |
|---|---|---|---|---|---|---|---|---|---|
| Date: | 1980, 1988, 2004 | | | Brown soil on limestone | | | | | |
| Locality: | Medvjeđak | | | | | | | | |
| Area: | 20 × 20 m (400 m$^2$) | | | Growing form: | | | | | |
| Elevation: | 825 m | | | Regular tall forest | | | +—insignificant | | |
| Coordinates: | N 44°53.683′ | | | | | | 1—1%–10% | | |
| | E 15°38.308′ | | | | | | 2—10%–25% | | |
| Bedrock: | Limestone | | | | | | 3—25%–50% | | |
| Exposition: | N | | | | | | 4—50%–75% | | |
| Inclination: | 10% | | | | | | 5—75%–100% | | |
| Cover: | | | | | | | | | |
| I. Tree layer: | | | | | | | | | |
| II. Shrub layer: | | | | | | | | | |
| III. Ground layer: | | | | | | | | | |
| Species: | Survey 1980 | Survey 1988 | Survey 2004 | Species: | | Survey 1980 | Survey 1988 | Survey 2004 | |
| I. | 95% | 90% | 85% | *Asplenium trichomanes* L. | | + | + | | |
| *Fagus sylvatica* L. | 5 | 5 | 5 | *Lunaria rediviva* L. | | + | + | 1 | |
| II. | 5% | 5% | 15% | *Actea spicata* L. | | + | + | | |
| *Sambucus racemosa* L. | 2 | | 2 | *Polistichum aculeatum* (L.) Roth. | | + | | | |
| *Rhamnus falax* Boiss. | 2 | 1 | 1 | *Sanicula euroaea* L. | | 1 | + | | |
| *Fagus sylvatica* L. | 1 | 1 | 1 | *Impatiens noli tengere* L. | | + | 1 | | |
| *Daphne mezereum* L. | 1 | + | + | *Glechoma hirsuta* W.K. | | + | | | |
| *Sambucus nigra* L. | | 1 | | *Lamium galeobdolon* Huds. | | | 1 | | |
| *Lonicera xylosteum* L. | | + | | *Festuca sylvatica* Huds. | | | 1 | | |
| *Corylus avellana* L. | | + | | *Scolopendrium vulgare* Sm. | | | 1 | | |
| *Euonymus verrucosa* Scop. | | + | | *Epilobium hirsutum* L. | | | + | | |
| *Crataegus monogyna* Jacq. | | + | | *Viola sylvestris* Lam. | | | + | | |
| *Abies alba* Mill. | | + | | *Solanum dulcamara* L. | | | + | | |
| *Lonicera alpigena* L. | + | | + | *Aconitum vulparia* Rchb. | | + | + | | |
| *Acer pseudoplatanus* L. | + | 1 | + | *Arum maculatum* L. | | + | + | | |
| *Fraxinus excelsior* L. | + | + | + | *Mercurialis perennis* L. | | + | + | + | |
| III. | 80% | 80% | 80% | *Milium effusum* L. | | + | + | | |
| *Asperula odorata* L. | 2 | 3 | | *Pulmonaria officinalis* L. | | + | 1 | + | |
| *Poligonatum multiflorum* (L.) All. | + | + | + | *Eupatorium cannabinum* L. | | + | + | + | |
| *Viola sylvatica* Lam. | + | | + | *Allium ursinum* L. | | | | + | |
| *Driopteris filix mas* (L.) Rich. | 2 | 1 | 2 | *Asarum europaeum* L. | | | | + | |
| *Circea lutetiana* L. | 1 | 2 | + | *Phyllitis scolopendrium* (L.) Newm. | | | | 1 | |

**Table 3.** *Cont.*

| | | | | | | |
|---|---|---|---|---|---|---|
| *Lamium orvala* L. | 1 | + | + | *Rubus hirtus* L. | + | + |
| *Cardamine savensis* Shultz. | + | 1 | + | *Acotinum licoctonum* L. | | + |
| *Mycelis muralis* (L.) Rchb. | + | + | + | *Anemone nemorosa* L. | | 1 |
| *Solanum dulcamara* L. | + | | + | *Stellaria nemorum* L. | | + |
| *Geranium robertianum* L. | + | + | 1 | *Cardamine enneaphyllos* (L.) Cr. | + | + |
| *Urtica dioica* L. | + | + | + | *Veratrum album* L. | | + |
| *Senecio nemorensis* L. | 1 | 1 | + | *Stellaria nemorum* L. | | |
| *Galium rotundifolium* L. | + | | 1 | *Euphorbia dulcis* L. | | + |
| *Brachypodium sylvaticum* (Huds.) R.B. | + | + | + | *Epilobium montanum* L. | + | |
| *Atropa belladonna* L. | + | | | *Lathyrus vernus* (L.) Bernh. | + | |
| *Galeobdolon luteum* Huds. | 1 | 1 | | *Doronicum austriacum* Jacq. | + | |
| *Fagus sylvatica* L. | + | 1 | + | *Carex pilosa* Scop. | 1 | |
| *Carex sylvatica* Huds. | + | + | | *Cardamine tryfolia* L. | + | |
| *Heracleum spondylum* L. | + | | | *Scrophularia nodosa* L. | + | |
| *Veronica montana* L. | + | + | | *Fragaria vesca* L. | + | |
| *Galeopsis tetrahit* L. | 1 | | | *Atropa belladona* L. | + | |
| *Athyrium filix femina* (L.) Roth. | + | 1 | 1 | *Neottia nidus avis* (L.) Rich. | + | |
| *Paris quadrifolia* L. | 1 | | | *Euphorbia carniolica* Jacq. | + | |
| *Salvia glutinosa* L. | + | 1 | | *Stachys sylvatica* L. | + | |
| *Euphorbia amygdaloides* L. | + | + | | *Hacquetia epipactis* (Scop.) DC. | + | |
| *Oxalis acetosella* L. | + | + | | *Prenanthes purpurea* L. | + | |
| *Melica uniflora* | | + | | *Cyclamen europaeum* Mill. | + | |

Table 4 and Figure 10 depict the process of natural regeneration through counts of young-generation trees and shrubs along three transects (see Figure 4) with a total surface area of 360 m$^2$ during the surveys conducted in 1998/2009/2018. A total increase in the number of plants per hectare was recorded, from 40,809 (1998) to 56,360 (2009), though this again decreased to 28,365 (in 2018). Most of these changes were in the representation of young beech plants, whose abundance increased fourfold from 3557 plants per hectare in 1998 to 12,694 plants per hectare in the second survey, with a further significant increase in the third survey to 22,731 (2018). In percentage share of the total number of young-generation trees and shrubs (in 1998), common beech accounted for 8.7%, as opposed to 22.6% in the second survey (2009) and 80% in the third survey (2018) (Figure 10), taking dominance over the new generations of trees. Considering the height structure of the young beech trees, the increase in abundance was most evident in the first two height classes to 60 cm (Table 4), and this was correlated with a reduction in the total number of old trees, opening of the canopy, and gaps that enable more intensive natural regeneration of beech. In terms of the quality of the young generation, most of the young beech plants were of poor quality with an uncertain future development. Given the current conditions of passive protection (as set by the legislation), it is not possible to stimulate regeneration or ensure the growth and development of the young beech trees. The ratio of remaining trees remained relatively constant over the years, unlike the large oscillations recorded in the shrub layer, with a significant reduction in the final survey year (2018).

**Table 4.** Abundance of young-generation beech trees, other trees, and shrubs per hectare (natural regeneration process) according to Dubravac et al. [19].

| Height Class | Beech | | | Other Trees * | | | Shrubs ** | | | Total | | |
|---|---|---|---|---|---|---|---|---|---|---|---|---|
| (cm) | 1998 | 2009 | 2018 | 1998 | 2009 | 2018 | 1998 | 2009 | 2018 | 1998 | 2009 | 2018 |
| to 30 | 1528 | 10,194 | 15,650 | 1556 | 1028 | 1136 | 28,167 | 36,194 | 3130 | 31,251 | 47,416 | 19,916 |
| 31–60 | 1417 | 1111 | 4016 | 1056 | 1306 | 621 | 5500 | 3917 | 111 | 7973 | 6334 | 4748 |
| 61–130 | 556 | 1139 | 1856 | 361 | 611 | 305 | 361 | 222 | 166 | 1278 | 1972 | 2327 |
| 131–150 | 28 | 139 | 378 | 56 | 28 | | | | 55 | 84 | 167 | 433 |
| 151–200 | 28 | 83 | 526 | 167 | 111 | 55 | | 56 | | 195 | 250 | 581 |
| 201–250 | | 28 | 250 | 28 | 83 | | | | | 28 | 111 | 250 |
| 251> | | | 55 | | | 55 | | | | | 110 | 110 |
| Total | 3557 | 12,694 | 22,731 | 3224 | 3167 | 2172 | 34,028 | 40,389 | 3462 | 40,809 | 56,360 | 28,365 |

Legend: * Other trees: sycamore maple, spruce, European ash; ** Shrubs: elderberry, daphnes, hazel, and others.

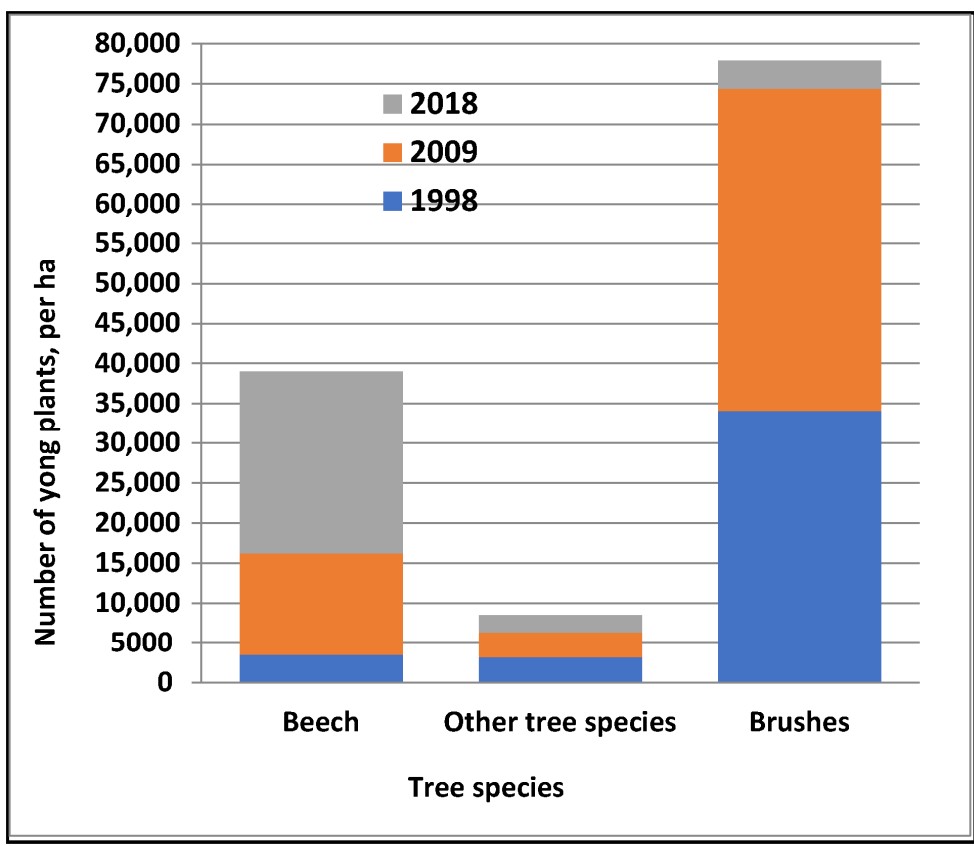

**Figure 10.** Abundance of young plants in each survey year.

The natural regeneration process was most affected by the opening of the canopy following the death of a dominant beech tree and the appearance of a gap (Figures 8 and 9). The surface area of the gap during the survey grew exponentially, from 124.7 m² in the first survey (1998) by 3.5 times to 512.2 m² in the second survey (2009) [19], and, finally, a sevenfold increase to 835.4 m² in the third survey (2018) (Figure 11).

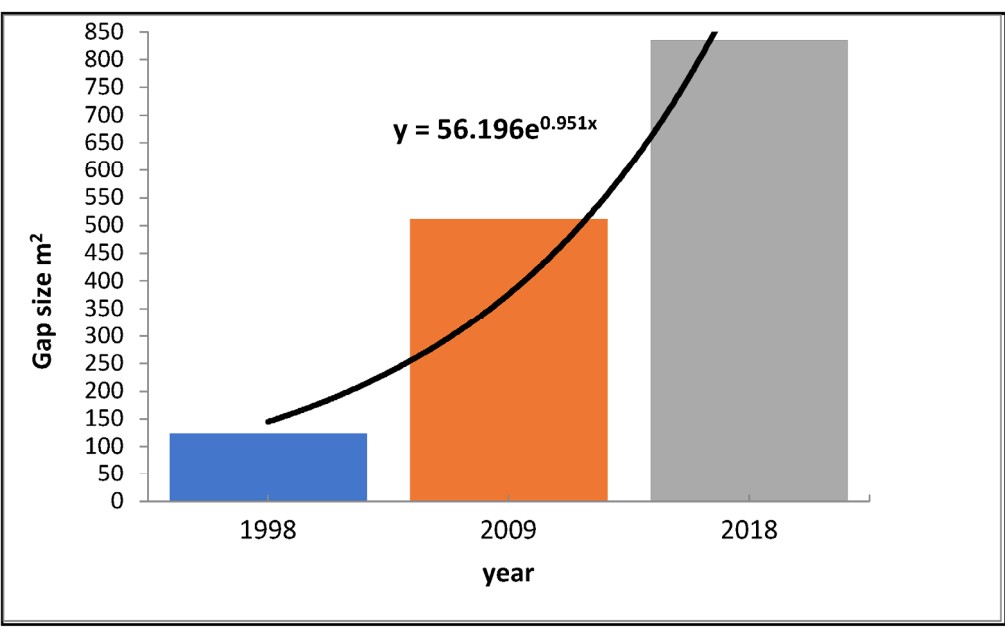

**Figure 11.** Gap size distribution in the Medvjeđak Forest Reserve.

## 4. Discussion

Protected areas are considered key elements in the conservation of biological diversity and in halting global losses to biodiversity [40–42]. In that context, the question arises of whether they need to be managed at all, and, if so, whether passive or active management is the best option. Discussions on this topic have given contrary conclusions [43–46]. Forest ecosystems are dynamic and constantly changing over time, and habitats and the biodiversity within those ecosystems will also change. The example of Plitvice Lakes National Park is an excellent indicator of the complexity of this issue.

Plitvice Lakes National Park was the first national park in Croatia, and it has been protected since 1949. On 26 October 1979, the lakes were also inscribed onto the UNESCO List of World Cultural and Natural Heritage. Throughout the broader protected area in the park, all procedures that would threaten the tributaries are strictly prohibited, while construction and use of water power for industrial purposes is permitted only if they have no effect on the natural flows and the level of water purity. In the narrow (strict) protected area, the 16 lakes with their waterfalls and tributaries must remain unchanged and in their natural composition. All procedures that would erode their natural composition are prohibited. The park area is divided into three zones, and, accordingly, into three protection levels:

(a)   Strict protection zone, 80.73%;
(b)   Moderate protection zone, 17.12%;
(c)   Use zone, 2.15%.

The area of the fundamental phenomenon encompasses the area of the lakes, waterfalls and the entire biodynamic process, the surface of aboveground flows of runoff water that belong to the gravitational catchment of the lakes, their tributaries, and the upper course of the Korana River, in which the biodynamic process of tufa formation has unfolded and will continue to unfold. The real and constant formation of tufa is the fundamental phenomenon of the Plitvice Lakes National Park, and it is an irreplaceable condition for the existence of the park. The phenomena of the Dinaric karst are found in a broader area within the park boundaries. Surface and subterranean karst forms appear in the form of *vrtače* (funnel-shaped depressions), sinkholes and sinking rivers, pits, caves, and ice caves. In addition to the aquatic ecosystems and karst phenomena, the forest ecosystem is another important determinant of the park, covering about 81% (24,044 ha) of the total land area. Therefore, conservation and regeneration of forest ecosystems is exceptionally important

for a balanced relationship in the protected area. Forests, as a natural vegetation form, have a strong impact on preserving the lakes. There is a mutual relationship between the lakes, watercourses, and forests. The forests prevent floodwaters and erosion, thereby enabling normal flow and accumulation of water, and therefore the areas under forest cover within the Plitvice Lakes water system are particularly important.

It is these forest ecosystems that demand stability and a balanced relationship among young, middle-aged, and old stands. In a protected area, such as a national park, this requires a shift in the paradigm and implementation of certain active management methods while leaving passive management behind. The current legislation concerning national parks in Croatia does not permit any form of forest management, though exceptions are permitted in certain conditions of sanitary harvest. In the development dynamics of beech forests in the national park, an important circumstance is the opportunity for natural regeneration, which can be achieved through active management measures, such as measures to regenerate old stands.

Due to its silvicultural properties, common beech tolerates various means of regeneration under the canopy of old trees, from shelterwood cutting over a large area to shelterwood cutting over small areas where one type of cutting (shelterwood) can be used or a combination of methods can be applied (such as shelterwood cutting with edge cutting). Shelterwood cutting in small areas can be used to create even-aged or uneven-aged stands, depending on the duration of the regeneration period for each part of the stand (section) being regenerated. The sum of regeneration periods needed to regenerate the entire stand (section) is called the overall regeneration period. In the procedure of regeneration in small areas with the aim of creating an even-aged stand, the regeneration period should not be longer than 20 years, while if the aim is to create an uneven-aged stand, then this can be longer than 20 years [1]. This is also in line with research on coastal beech forests in the Učka Nature Park in Croatia, which has shown that it is possible to achieve even-aged management using shelterwood cutting in groups through four cuts over a regeneration period of 30 years, which is aligned with the protective forest function and ensures that sustainable forestry management is achieved. In the national park area, this regeneration method gives the best results as it allows for the shaping of a structurally diverse stand that will satisfy the general functions of beech forests. Research to date in this area has shown that regeneration using shelterwood cutting in small areas (in clumps) is the regeneration method that is closest to nature, and it can also be applied in special purpose and protected forests [19,24,47].

Furthermore, forest care is directed towards developing stands in line with the natural laws by optimally considering the habitat conditions, biological properties, and ecological demands of the tree species under management. In this way, it is possible to keep the stand in optimal structural conditions, where stability, productivity, and biological diversity are the fundamental indicators of the optimal condition.

These measures could be built into the management mechanisms of national parks (management plans) with the aim of achieving successful regeneration and a favourable stand structure. This in no way lessens the value of the protected area, as protected areas are very important in nature conservation and environmental protection policies focused on the protection of biodiversity [48,49]. However, the protected area's goal has evolved [50] to deliver many ecological, social, and also economic benefits [51].

Though the research was conducted over a relatively short time period for a forest ecosystem (1980–2018, nearly 40 years), the methods used and results obtained in this study indicate a better understanding of the structural dynamics and natural regeneration processes of pure beech stands under passive management conditions. We believe that our results indicate the possibility for a different approach in the management of the protected area of the national park. Namely, the stability of forests can be achieved by encouraging natural regeneration. The facts for that claim can be confirmed in Figure 10 and Table 4. In this case, it is necessary to take advantage of the possibility of natural regeneration for the dominant species of common beech. Also, in Table 3, you can see the slight representation

of common beech in the ground layer in 2004. Of course, we do not consider that it is necessary to economically evaluate the forests in the protected area. On the other hand, the ecological roles come to be valued to a greater extent if they would enable limited management.

In line with the research conducted in the old-growth forests of Slovakia [9], and based on this high wood stock in the present study with a small number of old, large, and overmature trees and many thin beech trees, it was established that the studied forest is in the decay phase with beech expansion.

The main drivers of development dynamics in protected forest ecosystems are canopy gaps that appear in stands due to the effects of abiotic and biotic factors [52–56]. They can be the projection area of a single tree crown to a clump of trees, or, following strong winds, they can cover an area of several hectares. It is in these gaps that the regeneration of forest stands begins, following changes in the ecological conditions, particularly light, temperature, and moisture, which enable the appearance of seedlings that form the regeneration core [47]. A similar study conducted by [12] showed that gaps of less than 500 m$^2$ were the dominant driving force for stand development. Therefore, we believe that the gradual opening of the canopy in small areas is the best form of natural regeneration, especially in zones where it would be acceptable in a protected area.

Regeneration cores can be the basis from which natural regeneration of beech stands will begin in those small areas. For this reason, it is important to know how many there are, how they are distributed in the stand space, their size and shape, and the appearance of the young generation forming them. Also important are the distribution of regeneration cores in the regenerated area, the manner in which they are freed from shade, and the rhythm and manner of how their expansion and connection change in line with the management aims for the stand. In the national park, this can be achieved through limited management only in some zones, which will enable and ensure the formation of young stands. According to Figures 8 and 9 and the measurements from that period, it is clear in which direction it is necessary to actively intervene if a balanced restoration is to be achieved.

With the aim of achieving scientifically-based plans for deciding on the future of forest stands in protected areas, particularly in conditions of climate change we are currently facing, it is necessary to constantly improve and expand monitoring methods using networks of permanent experimental plots (for monitoring) in all protected forest areas. Nor should we ignore the opportunities that modern computer models can provide in properly depicting this gained knowledge. Modern methods and remote sensing data (such as satellite imagery, LiDAR, and drones) can be used for the characterisation of forest dynamics in large areas in protected areas.

## 5. Conclusions

The research to date confirmed in this survey shows that natural regeneration needs to unfold in initial regeneration cores, in small-sized canopy gaps, and in clumps. This approach is the closest to nature's means of regeneration, particularly in protected forest ecosystems. In abiding by the existing laws and regulations in areas that are not under strict protection regions, we need to facilitate positive natural processes that are already unfolding in the forest ecosystems. These activities are a fundamental means to optimise the priority forest functions in the protected area of a national park, in which three principles are emphasised: the best production forest is also the biologically most stable forest that provides the highest general functions. In that sense, active management contributes to the natural balance, making the forest ecosystem more resilient to unfavourable external factors, particularly the increasingly frequent climate extremes. Third, silvicultural treatment should be a necessary tool for those who manage the forest ecosystem. Under passive protection conditions, the beech stands take on a structure in which the vitality, the regeneration potential, and the appearance of new generations decline, which consequently has a negative effect on their overall sustainability. If they cannot break out of the shade of old tree crowns, the young beech generation becomes deformed and incapable of forming

a new generation of forest stands. We believe that this is one of the ways of conserving the forest ecosystem in response to the present-day challenges of how to stimulate natural regeneration in a way that would bring greater stability and resilience for the national park forests in tolerating and surviving the more pronounced climatic extremes.

**Author Contributions:** Conceptualisation, D.B. and T.D.; methodology, T.D. and D.B.; software, T.D.; validation, T.D. and D.B.; formal analysis, T.D.; investigation, T.D., Ž.Š., and D.B., resources, T.D., S.V., and D.B.; data curation, T.D.; writing—original draft preparation, D.B.; writing—review and editing, T.D. and D.B.; visualisation, T.D. and S.V.; supervision, Ž.Š.; project administration, D.B. and R.R.; funding acquisition, T.D. and R.R. All authors have read and agreed to the published version of the manuscript.

**Funding:** This research received no external funding.

**Data Availability Statement:** The data presented in this study are available on request from the corresponding author. The data are not publicly available due to continuation of research.

**Acknowledgments:** The authors thank the employees of the public institution Plitvice Lakes National Park, and especially Nikola Magdić.

**Conflicts of Interest:** The authors declare no conflicts of interest.

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
