# Peer review of "The Dynamics of Stand Structure Development and Natural Regeneration of Common Beech (Fagus sylvatica L.) in Plitvice Lakes National Park"

_forests, doi:10.3390/f15020357_

Round 1
Reviewer 1 Report
Comments and Suggestions for Authors
Dear editors and authors,
The manuscript is very interesting by analyzing the dynamics of stand structure in a forest reserve with a focus on regeneration of common beech. Although beech regeneration is generally ensured, the study highlights the quality of regeneration. I would change the title in order to emphasize the focus on natural regeneration as a result of stand structure development under a passive management. I don’t find the word “possibilities” from title too specific, it is too general. I think also that the discussion section should be reconsidered, it refers less to the results of the study and more to general information.
For improving the manuscript, some comments on the style and minor content remarks will be given below.
Line 11: “The paper investigates”… My question is: the paper or the authors investigate?
Line 18-19: “The distribution of the number of trees of the first two measurements (1998-2009) indicates a distribution with pronounced right asymmetry.” What kind of distribution do you refer to?
Lines 29-40: The text requires citations.
Line 74: Please, replace [22, 18] with [18, 22].
Lines 80-82: “Further, an important part of the research is focused on the possibilities of satisfactory natural regeneration within national parks”. Please, be more specific and replace “possibilities”.
Lines 116-124: The text requires citations.
Figure 1: In the figure, the text is too small and cannot be read.
Line 144: “(60 x 60 m)” in other cases (on line 154, on line 159) is written without spaces.
Lines 145-146: “the elevation of terrain elevations” this text needs to be reformulated.
Line 149: You should delete a comma before “[40]”.
Lines 155, 175, 183, 242, 245: “according to [18]” – a word should be added before citation.
Figure 4: The position of the value 60 m is wrong. The values 60 m should be above the line in both cases.
Line 189: Please, replace “indicate” with “indicates”.
Line 190: It is correct to obtain a pronounced right asymmetry by DBH classes instead a left asymmetry in an even-aged stand? Could you please, verify the skewness for DBH? It is positive or negative?
Line 205 and in other places: you should discuss the stand basal area in m2ha-1, not m2.
Line 208-210 and in other places: you should discuss the stand volume in m3ha-1 (or m3/ha), not m3 and you should keep the same way.
Line 212: “mature trees of common birch” – are you sure? It is birch?
Table 1: The symbols are not explained, the units need corrections.
Figure 5: The distribution show a left asymmetry!! Why do you choose DBH classes of 5 cm? The values from horizontal axis cannot be read.
Figure 6: Do not use decimals on vertical axis! The values from horizontal axis cannot be read.
Figure 7: You should find a better place for legend.
Line 224: “The average surface of the horizontal canopy projections was 53.67 m2” – here canopy or crown?
Line 235: Canopy or crown?
Line 236: Replace m2 with m2.
Line 259: Add a comma in number 3557.
Table 3: Introduce a comma to delineate thousands in the values of the abundance. Lines form the table should be edited.
Figure 11: Delete the dot after the year on horizontal axis. There are too many figures in the manuscript. You can include 2 or more charts in the same figure.
Discussion is not totally fit to the results. In fact, you discuss your results just a little bit, from line 366 to 399. More, from lines 289 to 365 you discuss general things too less related to your results, better for other sections (introduction or material). As such, I think that discussion section should be reconsidered.
Line 382: “national regeneration” is correct? Or do you refer to natural regeneration?
Reviewer 2 Report
Comments and Suggestions for Authors
The stability of forest ecosystems depends on the interaction of abiotic and biotic factors, that either directly or indirectly affect the morphology and dynamics of stands. The abiotic and biotic factors in forest ecosystems cause various disturbances. Thus, the wind, fungi, and insects through interaction cause the fall of a dead standing tree, initiating the formation of canopy gaps. These openings are considered as basic initiators of virgin forest dynamics in the European moderate zone (Korpel 1989, Diaci et al. 2003, Nagel et al. 2006). The newly formed openings are the places where the natural regeneration of a forest stand begins. It is there where the dynamic process of forest stand regeneration takes place (Watt, 1923).
In the manuscript submitted by Dubravac et al., the structural characteristics, regeneration processes, growth, development and survival of the young generation of common beech (Fagus sylvatica L.) were examined based on three periodic measurements (1998, 2009, 2018) made in the Plitvice Lakes National Park in Croatia.
In the literature on the subject, the authors addressed this problem. In this respect, the work is not original. The manuscript is an extension of previously published research from this area (https://doi.org/10.15177/seefor.13-10). A significant part of the published results [18] was reproduced in the manuscript submitted for review. This applies to research carried out in 1998 and 2009. Authors should add a reference to results already published. For example, in Table 3 for the 1998 and 2009 results, such a reference should be added [18].
In this review, I offer a few suggestions as to where certain points can be elaborated upon or revised in the manuscript.
Detailed comments:
1. Lines 36-38: Add reference to the statement: “Fortunately, beech forests are in relatively good health with good resilience to environmental pollution in comparison with common fir and pedunculate oak that are more susceptible to sudden ecological changes.”
2. Lines 107-108: Add reference: “…Köppen classification […]…”
3. There is no reference to Figure 2 in the text.
4. Lines 188-191: Confirm this statement with statistical analysis: “The distribution of tree numbers (Figure 5) in the first two measurements (1998–2009) indicate a normal (Gaussian) distribution of tree numbers with a pronounced right asymmetry that follows the relatively constant number of trees by DBH classes. Trees are grouped around the mean DBH (40 cm, Table 1). The most recent measurement interval (2018) shows a bimodal Gaussian distribution, with the first peak between 7.5 to 52.5 cm, and the second from 52.5 to 92.5 cm, with a significantly higher frequency of trees with a greater DBH classes”.
5. Lines 182-183: The sentence requires stylistic correction: “…their syntaxonomy position and sociological association of individual species are taken according [34].”
6. Line 206; 209; 225; 229; 233; 234; 236; 237; 238; 282; 283: To avoid self-plagiarism, please add a reference to your publication [18]. Please check the entire manuscript and add references to previously published results. For example:
- line 206:”…(2009) was 45.68 m2 [18], for an ….”
- line 209: „…803.07 m3/ha (2009) [18] and…”
- line 225: „…was 53.67 m2 with a range from 5.94 to 158.36 m2 [18]”.
- line 145: “…145 m2 [18]”
- line 234: “…55.25 m2 [18]”
- line 236:”…69.27 m2 [18]”
- line 238: 1512.89 m2 (+20.65 m2) [18].”
- line 282: “…124.7 m2 [18]…”
- line 283:”… 512.2 m2 [18]…”
7. Line 212: Is “common birch”. Should be: “common beech”
8. Table 1: Consider removing the unit from the first line (N/ha; G/ha; V/ha), because there is repetition in the second line.
9. Table 1. Add explanations of the symbols used below table (table footer). Tables must be auto-explicative.
10. Figures 5 and 6. Improve the quality of Figures 5 and 6 (X-axis is illegible).
11. Figure 7: Please correct the unit formatting on the Y axis.
12. Table 3: Please add a reference [18] to the results obtained in 1998 and 2009 (published).
13. Lines 400-419: Conclusions. In the end, the novelty of the work has to be emphesized.
14. Reference Lines 432-550: References not prepared in accordance with the requirements journal Forests (Forests | Instructions for Authors (mdpi.com). For example: Line: 479, 521, 533
Author 1, A.B.; Author 2, C.D. Title of the article. Abbreviated Journal Name Year, Volume, page range.
Reviewer 3 Report
Comments and Suggestions for Authors
General remarks of the reviewer
The work carried out interesting research on the structural features, regeneration processes, growth, development and survival of the young generation of common beech in a forest reserve in the Plitvice Lakes National Park in Croatia.
The following chapter requires some clarification:
3. Results
The paper presents the distribution of the number of trees and volume in DBH classes at the level of statistical generality. It would be interesting to approximate empirical distributions with selected theoretical distributions (e.g. normal, log-normal, Weibull, Gaussian mixture, SB Johnson).
Correct incorrectly determined numerical values in the text and table 1.
Figure 5,6: Make the BHD classes scaling more clear.
Figure 7: BHD classes must be strictly closed.
Technical Notes
Eliminate logical errors and typos.
If possible, make the figures clearer.
The description of the literature item needs to be corrected as required by the publisher: articles, books and other sources - italics of journal titles, year in bold, correct pages of journals and the access link and date of access in English. According to MDPI standard.
Details in the attached manuscript.

Comments on the Quality of English Language
Minor editing of English language required.
Round 2
Reviewer 1 Report
Comments and Suggestions for Authors
Dear authors and editor,
The revision was satisfactory done and the comments were mostly considered. Below you will find some suggestions for a minor revision of the manuscript.
Line 19: Please, replace “investigates” with “investigate”.
Line 81: “[8-9] and [2, 10-12]” could be cited as [2, 8-12].
You have used m2ha-1 and m3/ha. Why did you not use m3ha-1 to keep the same manner for units in the article? It is the same, but why should we write in two different ways?
Table 1: I’m not sure that you should include “ha” in N/ha, G/ha, and V/ha. Usually N, G, and V are by definition expressed per hectare. Why d1.30 and not the same DBH as in the text of the article?
Lines 269-270: The legend should refer to symbols used in table, so you should explain as follows: N – number of trees per hectare, G – stand basal area, V – stand volume, and average values of tree characteristics (DBH – diameter at breast height, h – tree height, g – tree basal area, v – tree volume).
Figure 7: On the vertical axis the unit should be corrected as m3ha-1.
The discussion was partially reconsidered. May be you should done more here according to my first comment reproduced below.
Discussion is not totally fit to the results. In fact, you discuss your results just a little bit, from line 366 to 399. More, from lines 289 to 365 you discuss general things too less related to your results, better for other sections (introduction or material). As such, I think that discussion section should be reconsidered.
Author Response
General concept comments:
- Line 19: Please, replace “investigates” with “investigate”.
Authors agree with suggestion:
Authors investigate……
- Line 81: “[8-9] and [2, 10-12]” could be cited as [2, 8-12].
Authors agree with suggestion:
of old growth beech forests [2, 8-12],……
- You have used m2ha-1and m3/ha. Why did you not use m3ha-1 to keep the same manner for units in the article? It is the same, but why should we write in two different ways?
Authors agree with suggestion:
We corrected units in table and also in text (m3ha-1 and m2ha-1 ).
- Table 1: I’m not sure that you should include “ha” in N/ha, G/ha, and V/ha. Usually N, G, and V are by definition expressed per hectare. Why d30and not the same DBH as in the text of the article?
Authors agree with suggestion:
Table 1. Basic structural characteristics
|
Measurement |
N |
G |
V |
DBH |
h |
g |
v |
- Lines 269-270: The legend should refer to symbols used in table, so you should explain as follows: N – number of trees per hectare, G – stand basal area, V – stand volume, and average values of tree characteristics (DBH– diameter at breast height, h – tree height, g – tree basal area, v – tree volume).
Authors agree with suggestion:
Legend: (N – number of trees per hectare, G – stand basal area, V – stand volume, and average values of tree characteristics (DBH – diameter at breast height, h – tree height, g – tree basal area, v – tree volume)
- Figure 7: On the vertical axis the unit should be corrected as m3ha-1.
Authors agree with suggestion:
We made correction on Figure 7.
- The discussion was partially reconsidered. May be you should done more here according to my first comment reproduced below.
Discussion is not totally fit to the results. In fact, you discuss your results just a little bit, from line 366 to 399. More, from lines 289 to 365 you discuss general things too less related to your results, better for other sections (introduction or material). As such, I think that discussion section should be reconsidered.
Authors agree with suggestion:
In the discussion section, we tried to improve what was requested.
Line 407-415
We believe that our results indicate the need for a different approach in the management of the protected area of the National Park. Namely, the stability of forests can be achieved by encouraging natural regeneration. The facts for that claim can be confirmed in Figure 10 and Table 4. In this case, it is necessary to take advantage of the possibility of natural regeneration for the dominant species common beech. Also, in table 3, you can see the slight representation of common beech in the ground layer in 2004. Of course, we do not consider that is necessary to economically evaluate the forests in the protected area. On the other hand the ecological roles come to be valued to a greater extent if they would enable limited management.

Reviewer 2 Report
Comments and Suggestions for Authors
Below, in red, are my comments on the authors' responses and the revised manuscript.
In the literature on the subject, the authors addressed this problem. In this respect, the work is not original. The manuscript is an extension of previously published research. Authors should add a reference to results already published.
Authors comment: With all due respect, the work is an original research with its continuation of the previously established monitoring. As we mentioned in Abstract and also in Materials and methods …. Initial research within the reserve began in 1980 as part of the international UNESCO project entitled “Man and Biosphere”, while vegetation mapping conducted as part of the IUFRO programme [33-34]…. Unfortunately, in forestry, we cannot only conduct experiments in the laboratory. It is necessary to monitor the state of nature (forest ecosystem) for a better understanding of certain changes and adaptation to different biotic and abiotic conditions. We add all the references. Moreover, we have too many self-citation which is justified objection of editor.
Reviewer's comment: The authors absolutely did not understand the reviewer's intentions. The addition of new results does not constitute a new research concept and in this respect the work is not original. I do not question field research at any point, on the contrary, I believe that it is an advantage, not a disadvantage, in forestry work. Therefore, the authors' comment is inappropriate.
Detailed comments:
Lines 188-191 (new lines 243-248) : Confirm this statement with statistical analysis: “The distribution of tree numbers (Figure 5) in the first two measurements (1998–2009) indicate a normal (Gaussian) distribution of tree numbers with a pronounced right asymmetry that follows the relatively constant number of trees by DBH classes. Trees are grouped around the mean DBH (40 cm, Table 1). The most recent measurement interval (2018) shows a bimodal Gaussian distribution, with the first peak between 7.5 to 52.5 cm, and the second from 52.5 to 92.5 cm, with a significantly higher frequency of trees with a greater DBH”.
Authors comment: In our opinion we showed distribution of the number of trees by DBH classes (Figure 5) and distribution of volume by DBH classes (frequencies of trees in 5 cm classes). Also we showed data in Table 1. If it is necessary to insert statistical analysis, should we remove some parts? The problem may arise if we exceed the limit of text and images.
Reviewer's comment: If authors are afraid of exceeding the limit of text and images in the manuscript, please include the statistical analysis in the supplement
Table 1: Consider removing the unit from the first line (N/ha; G/ha; V/ha), because there is repetition in the second line.
Authors comment: We tried to arrange table according to reviewer 1 and reviwer 2
Reviewer's comment: I still think that the unit in the first line is redundant and should be removed (N/ha; G/ha; V/ha).
Table 3: Please add a reference [18] to the results obtained in 1998 and 2009 (published).
Authors comment: Authors add reference in the text
Reviewer's comment: Also add a reference to table 3. Especially since ¾ of the results from table 3 have already been published. Tables must be auto-explicative.
Reference Lines 432-550 (new lines 681-838): References not prepared in accordance with the requirements journal Forests (Forests | Instructions for Authors (mdpi.com). For example: Line: 479, 521, 533
Authors agree with suggestion
Reviewer's comment: References not prepared in accordance with the requirements journal Forests e.g. item: 41, 45, 48.

Author Response
General concept comments:
- Reviewer's comment: The authors absolutely did not understand the reviewer's intentions. The addition of new results does not constitute a new research concept and in this respect the work is not original. I do not question field research at any point, on the contrary, I believe that it is an advantage, not a disadvantage, in forestry work. Therefore, the authors' comment is inappropriate.
Authors comment:
Although we do not agree with the statement, we respect the opinion in the context of the work.
- Detailed comments:
Lines 188-191 (new lines 243-248) : Confirm this statement with statistical analysis: “The distribution of tree numbers (Figure 5) in the first two measurements (1998–2009) indicate a normal (Gaussian) distribution of tree numbers with a pronounced right asymmetry that follows the relatively constant number of trees by DBH classes. Trees are grouped around the mean DBH (40 cm, Table 1). The most recent measurement interval (2018) shows a bimodal Gaussian distribution, with the first peak between 7.5 to 52.5 cm, and the second from 52.5 to 92.5 cm, with a significantly higher frequency of trees with a greater DBH”.
Authors agree with suggestion:
Line 230
|
measurement years |
1998. |
2009. |
2018. |
|
|
|
|
|
|
Mean |
39,10 |
41,11 |
46,83 |
|
Standard Error |
0,928867 |
0,983053 |
2,09356 |
|
Median |
38,00 |
40,50 |
44,30 |
|
Mode |
37,00 |
35,50 |
39,75 |
|
Standard Deviation |
16,12 |
16,77 |
17,64 |
|
Sample Variance |
259,7007 |
281,2206 |
311,1924 |
|
Kurtosis |
0,287316 |
0,141323 |
-0,60844 |
|
Skewness |
0,671723 |
0,587233 |
0,258718 |
|
Range |
89 |
89,5 |
78,05 |
|
Minimum |
11,00 |
11,00 |
10,90 |
|
Maximum |
100,00 |
100,50 |
88,95 |
|
Sum |
11768 |
11963 |
3324,853 |
|
Count |
301 |
291 |
196 |
- Reviewer's comment: I still think that the unit in the first line is redundant and should be removed (N/ha; G/ha; V/ha). Table 1: Consider removing the unit from the first line (N/ha; G/ha; V/ha), because there is repetition in the second line.
Authors agree with suggestion:
Table 1. Basic structural characteristics
|
Measurement |
N |
G |
V |
DBH |
h |
g |
v |
- Reviewer's comment: Also add a reference to table 3. Especially since ¾ of the results from table 3 have already been published. Tables must be auto-explicative.
Authors agree with suggestion:
Table 3. Abundance of young generation beech trees, other trees and shrubs per hectare (natural regeneration process) according to Dubravac et al. [19].
- Reviewer's comment: References not prepared in accordance with the requirements journal Forests e.g. item: 41, 45, 48.
Authors comments:
Forests template:
- Author 1, A.B.; Author 2, C.D. Title of the article. Abbreviated Journal Name Year, Volume, page range.
- Cent, J.; Grodzińska-Jurczak, M.; Pietrzyk-Kaszyńska, A. Emerging multilevel environmental governance – A case of public participation in Poland. Nat. Conserv. 2014, 22(2), 93-102.
- Schreiber, E.S.G.; Bearlin, A.R.; Nicol, S.J.; Todd, C.R. Adaptive management: a synthesis of current understanding and effective application. Manag. Restor. 2004, 5, 177-182.
- Gaston, K.J.; Jackson, S.F.; Cantú-Salazar, L.; Cruz-Pinón, G. The ecological performance of protected areas. Rev. Ecol. Evol. S. 2008, 39(1), 93–113. https://doi.org/10.1146/annurev.ecolsys.39.110707.173529